# Plant Responses of Maize to Two *formae speciales* of *Sporisorium reilianum* Support Recent Fungal Host Jump

**DOI:** 10.3390/ijms242115604

**Published:** 2023-10-26

**Authors:** Lukas Dorian Dittiger, Shivam Chaudhary, Alexandra Charlotte Ursula Furch, Axel Mithöfer, Jan Schirawski

**Affiliations:** 1Department of Genetics, Matthias Schleiden Institute, Friedrich Schiller University Jena, Philosophenweg 12, 07743 Jena, Germany; lukas.dorian.dittiger@uni-jena.de (L.D.D.); shivam.chaudhary@uni-jena.de (S.C.); 2Department of Plant Physiology, Matthias Schleiden Institute, Friedrich Schiller University Jena, Dornburgerstr. 159, 07743 Jena, Germany; alexandra.furch@uni-jena.de; 3Research Group Plant Defense Physiology, Max Planck Institute for Chemical Ecology, Hans-Knöll-Straße 8, 07745 Jena, Germany; amithoefer@ice.mpg.de

**Keywords:** smut fungus, maize, transcriptome, host jump, defense response, GO analysis, RT-qPCR, jasmonic acid, salicylic acid, phytohormones

## Abstract

Host jumps are a major factor for the emergence of new fungal pathogens. In the evolution of smut fungi, a putative host jump occurred in *Sporisorium reilianum* that today exists in two host-adapted *formae speciales*, the sorghum-pathogenic *S. reilianum* f. sp. *reilianum* and maize-pathogenic *S. reilianum* f. sp. *zeae*. To understand the molecular host-specific adaptation to maize, we compared the transcriptomes of maize leaves colonized by both *formae speciales*. We found that both varieties induce many common defense response-associated genes, indicating that both are recognized by the plant as pathogens. *S. reilianum* f. sp. *reilianum* additionally induced genes involved in systemic acquired resistance. In contrast, only *S. reilianum* f. sp. *zeae* induced expression of chorismate mutases that function in reducing the level of precursors for generation of the defense compound salicylic acid (SA), as well as oxylipin biosynthesis enzymes necessary for generation of the SA antagonist jasmonic acid (JA). In accordance, we found reduced SA levels as well as elevated JA and JA-Ile levels in maize leaves inoculated with the maize-adapted variety. These findings support a model of the emergence of the maize-pathogenic variety from a sorghum-specific ancestor following a recent host jump.

## 1. Introduction

Host jumps are the main causes for the emergence of new fungal plant diseases with the potential to threaten food production. New fungal diseases can emerge when a fungal pathogen of a certain plant suddenly gains the capacity to infect a different plant and thus changes its host. One prominent example is the appearance of wheat blast caused by the known rice pathogen *Magnaporthe oryzae* that was first found in the mid-1980s in Brazil [1] and that caused a substantial loss of wheat production in Bangladesh in 2016 [2,3]. Events of host jumping during evolution of fungal strains can be detected via phylogenomic analysis, comparing the phylogenetic tree of plant-pathogenic fungal strains with that of their host plants [4]. Phylogenomic analyses showed that smut fungi speciated at the time of speciation of their host plants, and that during the following evolution of the species complex, host jumps occurred several times [5,6].

One example of a proposed relatively recent host jump among the smut fungi is *Sporisorium reilianum*, a species that exists in two host-adapted varieties or *formae speciales* [7,8]. *S. reilianum* f. sp. *reilianum* is a pathogen of sorghum and sudangrass, while *S. reilianum* f. sp. *zeae* is a pathogen of maize [9]. Phylogenetic analyses show a closer relationship of *S. reilianum* with sorghum pathogens [6], supporting the notion that the maize-infecting strains of *S. reilianum* may have evolved from the sorghum-infecting strains [10], and that thus sorghum might have been the original host of *S. reilianum*. Based on genetic differences, the divergence time of the two *formae speciales* of *S. reilianum* has been estimated to be about one million years ago [11], showing that the two varieties do not regularly interbreed in nature.

Plant infection by *S. reilianum* requires germination of the diploid teliospores that show tetrapolar heterothallism. Tetrapolar heterothallism is caused by the existence of two required mating type loci, *a* and *b*, that reside on different chromosomes [12] and each occur in different alleles. The *a* locus occurs in three alleles and encodes the genes for a three-way pheromone/pheromone receptor system. Each *a* locus contains one pheromone receptor gene and two genes encoding different pheromones that are respectively recognized by the pheromone receptors encoded on the other two *a* alleles [13]. The *b* locus occurs in at least six alleles and encodes different versions of a heterodimeric transcription factor that is only active if the two subunits are encoded on different *b* alleles [13]. Spore germination results in lemon-shaped sporidia that can multiply mitotically. Sporidia with differences in both *a* and *b* mating type loci can recognize each other and fuse, forming a dikaryotic filament that is infection-competent [12,13]. Seedling plants can be easily infected by leaf whorl inoculation with sporidial mixtures [14], where the dikaryotic filaments form on the leaf surface and penetrate the leaf through an infection structure with the help of a well-balanced mix of different cell-wall-degrading enzymes. Both *formae speciales* of *S. reilianum* are biotrophic plant pathogens that can penetrate and spread in leaves of both sorghum and maize [9]. On sorghum, the maize-specific variety will encounter heavy plant defense reactions such as increased ROS production, the formation of callose depositions, and the generation of phytoalexins that prevent further spread of the fungus. The sorghum-adapted variety will be able to enter the vasculature, spread through the nodes into the apex of the plant, and cause head smut disease upon appearance of inflorescences. On maize, both *formae speciales* can multiply well in the leaves, travel through the vasculature, and grow within the plant until appearance of the inflorescences. However, while the maize-adapted variety is able to replace floral organs with fungal spore-filled sori on maize, the sorghum-adapted variety seems to reach the maize inflorescences only occasionally and then does not lead to the formation of sori or spores [9].

Different races of *S. reilianum* f. sp. *reilianum* have been described, showing virulence on particular sorghum cultivars while being nonvirulent on others [14,15]. This type of plant–fungus interaction can be nicely explained by the gene-for-gene hypothesis [16] that predicts the existence of gene pairs in host and pathogen, where the host gene encodes a resistance protein (R) that recognizes a pathogen protein encoded by an avirulence (AVR) gene. Recognition would lead to plant defense and resistance against the pathogen. Absence of either the R gene in the plant or the AVR gene in the pathogen would lead to susceptibility of the plant and virulence of the pathogen. Interestingly, only one race *of S. reilianum* f. sp. *zeae* has been found so far [8,17], supporting the notion that host specificity of the two *formae speciales* of *S. reilianum* is determined differently in sorghum and maize [9].

To better understand the plant processes involved in host selection of *S. reilianum*, we recently analyzed the plant responses of sorghum seedling leaves colonized by either one of the two *formae speciales* [18]. This analysis showed distinct plant responses to each *forma specialis*. Only the presence of *S. reilianum* f. sp. *zeae* induced a plethora of defense reactions in sorghum. In addition, we found indications for a potential change in the lipid composition of the plant. In contrast, the presence of *S. reilianum* f. sp. *reilianum* in sorghum led to upregulation of genes involved in detoxification of cellular oxidants, of genes involved in the unfolded protein response, and of genes potentially modifying the cuticle wax and lipid composition of the plant [18]. Here we complement this analysis by determining the plant response of maize seedling leaves to the same *formae speciales* of *S. reilianum*. Interestingly, we found strong indications for active plant defense responses and an activated unfolded protein response in the endoplasmic reticulum for both varieties. Both *formae speciales* also induced variety-specific plant genes. Inoculation with *S. reilianum* f. sp. *zeae* induced, among others, gene expression of chorismate mutases and oxylipin biosynthesis enzymes. Congruently, we found that salicylic acid (SA) reporter genes were significantly less expressed in leaves inoculated with the maize- than with the sorghum-adapted variety. In addition, concentrations of JA and the active signaling compound JA-Ile were increased, and SA concentration decreased in maize leaves inoculated with *S. reilianum* f. sp. *zeae*. These plant responses indicate that *S. reilianum* f. sp. *zeae* is not a well-adapted maize pathogen, which supports the hypothesis that the existence of SRZ is the result of a relatively recent host jump event of *S. reilianum*.

## 2. Results

To learn more about the plant response of maize towards the two host-specific *formae speciales* of *S. reilianum*, we reanalyzed the transcriptome data of a plant infection experiment [9]. During this experiment, three samples were generated by pooling RNA of three independent replicates. The three samples were maize plants inoculated with water (H_2_O), maize plants inoculated with a mixture of two mating-compatible strains of *S. reilianum* f. sp. *reilianum* (Zm-SRS), and maize plants inoculated with a mixture of two mating-compatible strains of *S. reilianum* f. sp. *zeae* (Zm-SRZ).

For reanalysis, the sequencing reads were mapped against the newest available version of the maize genome assembly [19]. Of the 40–70 million reads that we obtained per sample, around 88% could be uniquely mapped on the maize genome and associated with the annotated gene models (Table 1).

To check whether the sequenced RNA pools represent a true average of gene expression, we repeated the infection experiment in a different greenhouse at a different location. We measured gene expression of twelve randomly selected maize genes using quantitative reverse transcriptase PCR (RT-qPCR). The measured gene expression measured using RT-qPCR corresponded well with the fragments per kilobase and million mapped reads (FPKM) values of the corresponding genes in the RNA sequencing dataset (Figure 1).

We performed pairwise comparisons of gene expression in the three samples based on their FPKM values and identified differently expressed genes. In the comparisons Zm-SRS vs. Zm-H_2_O and Zm-SRZ vs. Zm-H_2_O, more differentially expressed genes were upregulated in the infected samples, suggesting that fungal presence induced maize gene expression. In comparison with Zm-H_2_O, more genes were upregulated in the compatible interaction of Zm-SRZ (1244 genes) than in that of Zm-SRS (939 genes; Figure 2A,B). The direct comparison between Zm-SRZ and Zm-SRS revealed that both *formae speciales* commonly induced a large set of genes. Interestingly, the number of genes specifically induced by SRZ is larger than that induced by SRS (Figure 2C).

To obtain insight into the potential function of the differentially expressed genes, we performed a GO term enrichment analysis of selected gene sets. Surprisingly, both *formae speciales* induced a large set of genes associated with GO terms describing plant defense reactions. Additionally, the presence of both varieties led to downregulation of genes associated with photosynthetic activity (Figure 3 and Appendix A). This suggests that both *formae speciales* are recognized by the plant as pathogens.

We next considered genes that were differentially expressed only in one of the two pathosystems. SRS-specifically induced genes were associated with protein biosynthesis. Additionally, genes associated with the GO terms systemic acquired resistance and fatty acid binding were upregulated and genes related to inositol-3-phosphate biosynthesis were downregulated in the direct comparison of Zm-SRS with Zm-SRZ (Figure 3 and Appendix A). This indicates that SRS needs to cope with additional plant defense reactions that are not induced by SRZ.

In contrast, SRZ-specifically induced genes were associated with the GO terms DNA replication, sterol binding and transport, chorismate mutase activity and glutathion transferase activity. In the direct comparison of Zm-SRZ vs. Zm-SRS, even more glutathione-S-transferase genes and genes involved in oxylipin biosynthesis were upregulated (Figure 3 and Figure S1). Oxylipins are precursors of the phytohormone jasmonic acid (JA), while chorismate mutases deplete precursors necessary for generation of salicylic acid (SA). Therefore, upregulation of genes encoding oxylipin biosynthetic enzymes and chorismate mutases may indicate that SRZ modulates the JA-SA balance that is known to regulate plant pathogen interactions.

To check whether the JA-SA balance is differently influenced by SRS and SRZ, we also determined expression of the SA reporter genes PR1 and PR5 in leaf samples from water- and *S. reilianum*-inoculated maize seedlings collected at 3 dpi. The SA reporter genes PR1 (Zm00001eb341580) and PR5 (Zm00001eb032600) were significantly less induced in maize leaves inoculated with SRZ than those inoculated with SRS (Figure 1), supporting a differential modulation of the hormonal balance in SRS- and SRZ-inoculated leaves.

We further determined phytohormone concentrations in leaf blades collected at 8 dpi. We observed a significant decrease in the concentration of SA and a significant increase in the concentrations of JA and JA-Ile. The concentration of the active signaling compound JA-Ile was increased more than ten times in SRZ- relative to mock-inoculated leaves (Figure 4). Since the level of the JA precursor compound 12-oxo phytodienoic acid (OPDA) was unchanged relative to mock-inoculated leaves at this time point, the increase in JA and JA-Ile concentration was mainly due to de novo biosynthesis rather than hormone activation from storage compounds. This supports that leaf colonization with SRZ induces JA biosynthesis and reduces the level of SA, as suggested by the transcriptome data.

We also measured the concentrations of abscisic acid (ABA) and of indole acetic acid (IAA). The ABA concentration was increased more than four times in leaf samples inoculated with SRZ, whereas the IAA concentration was significantly reduced relative to mock-inoculated leaf samples (Figure 4). In contrast to SRZ, the concentrations of the measured phytohormones and phytohormone precursors (SA, cis-OPDA, JA, JA-Ile, ABA, and IAA) were unchanged in leaf samples inoculated with SRS (Figure 4). Based on these results, the key difference between the two *formae speciales* seems to be that only SRZ can manipulate the maize phytohormone system.

## 3. Discussion

In this study, we analyzed the transcriptome of maize seedling leaves inoculated with either the non-adapted *forma specialis* (SRS) or the adapted *forma specialis* (SRZ) of *S. reilianum*. Phenotypically, maize leaf inoculation with any *forma specialis* leads to mild symptoms on the leaves visible as chlorotic spots [9]. Both *formae speciales* are clearly recognized by the plant as unwanted pathogens, since gene expression of various defense genes, like chitinase genes and genes for the biosynthesis of antifungal secondary metabolites, is induced. In addition, the plant seems to respond to an induced accumulation of reactive oxygen species (ROS), presumably directed against the pathogen, by upregulation of genes involved in H_2_O_2_ catabolism. In congruence with the phenotypic leaf response, expression of genes required for photosynthesis is reduced. This situation is very different from the plant responses that both fungal varieties cause on sorghum. On sorghum, SRZ causes expression of a plethora of different defense genes [18], which corresponds well to the strong plant response visual on SRZ-inoculated leaves at three to seven days post-inoculation [20]. In contrast, except for genes coping with the consequences of ROS accumulation, plants inoculated with SRS do not show induction of the SRZ-induced defense genes, suggesting that at 3 dpi, SRS successfully prevents upregulation of further defense responses.

The maize smut pathogen *Ustilago maydis* is a close relative of *S. reilianum*. Its original host is thought to be teosinte, the progenitor plant of maize, suggesting that it never underwent a host jump event in its evolution. In the *U. maydis*–maize pathosystem, it is known that the fungus successfully represses initially induced defense responses within a couple of hours after plant penetration [21]. The very efficient downregulation of early induced defense gene expression after penetration shows that the fungus is highly adapted to its host plant. Like *U. maydis* on maize, SRS on sorghum does not show defense gene expression at 3 dpi [18], suggesting a similar long-time adaptation. In contrast, SRS and SRZ cause comparable defense responses on maize, indicating that the time for host adaptation has been minimal so far.

Assuming that the common ancestor of SRS and SRZ was a sorghum-adapted pathogen very similar to SRS, it is interesting to look at the behavior of SRS on maize. SRS can penetrate, multiply, and spread in seedling leaves of *Z. mays* cv. ‘Gaspe Flint’, can grow into the nodes, and can be found near the apical meristem of the plant [9]. In comparison with SRZ, maize systemic colonization by SRS is not as efficient, since the total fungal mass of SRS relative to the plant material is much smaller than for SRZ [9]. Although SRS is not able to form spores on *Z. mays* cv ‘Gaspe Flint’, it can induce phyllody in the inflorescences of inoculated plants [20]. This shows that SRS is already quite capable of coping with induced defense responses of maize, possibly needing only relatively small adjustments to successfully complete its life cycle.

Our analysis indicates that the required small adjustments may include the capability to modulate the concentration of plant hormones in favor of the fungus. SA is the main defense hormone against biotrophic fungi [22]. Common strategies of biotrophic fungi rely on downregulation of SA concentration and/or signaling, for example by depletion of SA precursor molecules, or by hijacking the mutually antagonistic relationship between SA and JA [23]. Inoculation of maize with SRZ leads to upregulation of chorismate mutase, an enzyme catalyzing the formation of prephenate from chorismate, thereby removing an essential precursor for SA biosynthesis. In addition, SRZ leads to upregulation of oxylipin biosynthesis. Oxylipins are essential precursors for JA biosynthesis. After the gene expression of chorismate mutase and oxylipin biosynthesis genes is selectively influenced, the level of JA in the plant tissue is expected to rise, which leads to a lowering of the SA level that is further lowered via precursor depletion. We could show that the effect on the phytohormone concentrations is persistent during leaf colonization since we detected reduced SA and elevated JA and JA-Ile concentrations even one week after plant penetration. Interestingly, we found a drastic increase in the concentration of ABA only in SRZ-inoculated leaves. ABA is thought to act negatively on SA signaling and positively on JA signaling, possibly via stabilization of DELLA proteins [24]. Therefore, an elevated ABA concentration might enhance the effect of SRZ on the SA-JA balance in the plant, further favoring the spread of SRZ.

In *U. maydis*, SA is known to be downregulated by the action of the secreted fungal chorismate mutase Cmu1 [25]. Upregulation of JA is achieved in *U. maydis* by the action of effectors interacting with TOPLESS, the corepressor of JA and IAA signaling [26,27]. Deletion of Tip1-Tip5 from the genome of the solopathogenic *U. maydis* strain SG200 leads to a slight reduction in virulence [27]. Both SRS and SRZ have homologs of these *U. maydis* effectors. However, the homologs are relatively weakly conserved, and their targets have not yet been elucidated. The homologs of Tip1 to Tip5 that occur in the cluster 6–10 of *S. reilianum* [12] can be deleted from the genome of SRZ without any effect on the virulence of SRZ on maize [28].

Measurement of reporter gene expression suggests that the SA level in SRS-inoculated maize leaves is increased relative to mock- or SRZ-inoculated leaves at 3 dpi. Congruently, we found ‘Systemic Acquired Resistance’ as the most significantly enriched GO term of genes that were upregulated in both the comparisons Zm-SRS vs. Zm-H_2_O and Zm-SRS vs. Zm-SRZ. Systemic acquired resistance is a mechanism of induced defense priming active in noninvaded tissue of an invaded plant. Elevated SA levels are required for systemic acquired resistance, and it is associated with upregulation of PR proteins [29]. In our experiments, both PR1 and PR5 are more highly upregulated in SRS- than in SRZ-inoculated leaves. Together with the fact that SRZ effectively interferes with the plant hormonal defense system, these differences in PR gene expression might indicate that only SRS is confronted with systemic acquired resistance when colonizing maize, possibly explaining its less efficient proliferation.

Why would a sorghum-adapted progenitor of SRZ that would be similar to today’s SRS be able to colonize maize? Plants are known to be equipped with an arsenal of nucleotide-binding site leucin-rich repeat (NBS-LRR) resistance proteins that directly or indirectly recognize proteins (AVRs) of their respective pathogens to induce defense. To date, only one NBS-LRR resistance protein of maize has been shown to contribute to resistance against a maize pathogen. The resistance protein RppC of maize was shown to directly bind to and recognize the AvrRppC effector protein of *Puccinia polysora*, the causative agent of Southern corn rust [30]. A systematic analysis revealed that the number of NBS-LRR receptor-encoding genes in maize is only half of that in sorghum despite the three-times-larger maize genome size [31]. Possibly, the small number of resistance receptors does not include one against smut fungi.

In an alternative scenario, it could be that resistance genes were already present before maize separated from sorghum. The progenitor of *S. reilianum* and *U. maydis* must have been able to effectively colonize the common ancestor of their hosts, since *U. maydis* and *S. reilianum* separated during speciation of maize and sorghum [6]. In the very long time that the progenitor of SRS and SRZ (proSR) evolved on sorghum, sorghum developed new resistance genes that proSR learned to cope with. Assuming that the maize resistance genes retained their original function, proSR would also be able to cope with the resistance genes of maize. Supporting this assumption is a recent finding that resistance genes against *Exserohilum turcicum* were conserved in both sorghum and maize [32]. To achieve a host jump of proSR to maize, the fungus would only need to develop slight adaptations. These adaptations could for example result in the fungal capacity to alter the plant’s hormone balance, thus generating a fungus with an enlarged host spectrum. During further adaptation to maize, the fungus will eventually lose genes that are directed towards sorghum resistance genes absent in maize. This would result in a maize pathogen unable to infect sorghum, like the current SRZ.

## 4. Materials and Methods

### 4.1. Experimental Design and Accession Numbers

Seven-day-old seedlings of *Zea mays* cv ‘Gaspe Flint’ were mock-inoculated with water (Zm-H_2_O), inoculated with the mating-compatible strains of the maize-adapted pathogen *Sporisorium reilianum* f. sp. *zeae*, SRZ1_5-2 and SRZ2_5-1 (Zm-SRZ) [20], or inoculated with the mating-compatible strains of the sorghum-adapted pathogen *Sporisorium reilianum* f. sp. *reilianum*, SRS1_H2-8 and SRS2_H2-7 (Zm-SRS) [20] and grown under greenhouse conditions in Aachen, Germany, as described [9]. Leaf pieces of a size of 3 cm from infected and mock-inoculated leaves were harvested at 3 dpi 1 cm below the inoculation site. For each treatment, the tissues of ten plants were pooled and the experiment was conducted in three independent biological replicates. RNA was isolated, and equal amounts of the replicates were pooled prior to RNA sequencing as described [9].

We submitted the transcriptome dataset to the NCBI Sequence Read Archive under the BioProject ID PRJNA1022446 with the BioSample accessions SAMN37607526, SAMN37607527, and SAMN37607528.

### 4.2. Transcriptome Data Analysis

The new transcriptome data analysis was performed with OmicsBox 3.0 [33]. The quality of the reads was assessed using FastQC [34], and low-quality reads were filtered using Trimmomatic [35]. The minimum average quality was set to 25 and the minimum read length to 36. The preprocessed reads were then mapped to the genome of *Zea mays* (Zm-B73-REFERENCE-NAM-5.0, [19]) using STAR [36]. Subsequently, a gene-level quantification was performed using the HTSeq software package (OmicsBox 3.0) and the expression was normalized to reads per kilobase per million mapped reads (RPKM) [37]. Differentially expressed genes were identified using the NOISeq software package (OmicsBox 3.0). Here, a count-per-million (CPM) filter of 1 was applied and the expression was also normalized to RPKM [38,39]. The mapping, gene-level quantification, and pairwise differential expression were otherwise performed with the default parameters of OmicsBox 3.0. For the GO term analysis of different sets of differentially expressed genes, the Fatigo software package of OmicsBox 3.0 was used to perform a Fisher’s exact test [40] with a false discovery rate-adjusted *p*-value of *p* < 0.05, yielding GO terms over-represented in the different gene sets in comparison with the reference genome of *Z. mays* [19].

### 4.3. Gene Expression Validation Via Real-Time PCR

To validate the RNA sequencing data, a RT-qPCR experiment was performed. For this, seven-day-old maize seedlings were inoculated with deionized water, with a mixture of the mating-compatible strains SRS1_H2-8 and SRS2_H2-7 (SRS), or with a mixture of the mating-compatible strains SRZ1_5-2 and SRZ2_5-1 (SRZ), and grown under greenhouse conditions (15 h day period at 28 °C, 9 h night period at 22 °C) in Jena, Germany. The inoculation was repeated thrice, and 20 plants each were used for each inoculation experiment. Leaf pieces of a size of 3 cm were collected 1 cm below the inoculation site at 3 dpi, frozen in liquid nitrogen, and stored at −80 °C. The leaf samples were ground to a fine powder using liquid nitrogen. Total RNA was isolated from 100 mg fine powder of each sample using ROTI^®^ Aqua-Phenol following the TRIzol^®^ protocol (Invitrogen, Thermo Fisher Scientific, Darmstadt, Germany). The concentration and integrity of RNA were measured with a NanoVue Spectrophotometer (Biochrom Ltd., Harvard Bioscience Inc., Berlin, Germany) and via 1% (*w*/*v*) agarose gel electrophoresis. To remove residual genomic DNA contamination from the sample, DNase treatment was performed using DNase l (Thermo Fischer Scientific, Darmstadt, Germany). DNA-free RNA was subsequently used for cDNA synthesis using a LunaScript^®^ RT SuperMix Kit (New England Biolabs, Frankfurt am Main, Germany). DNase treatment and cDNA synthesis were performed according to the manufacturer’s instructions. The primer efficiency of all the primer pairs used in the study was calculated and was between 90 and 100 (Table 2). A real-time PCR was conducted in a CFX 96 Thermal cycler (Bio-Rad Laboratories, Hercules, CA, USA), and the GAPDH gene was used as a reference gene [41]. The reaction was carried out in 96-well plates under cycling conditions of 95 °C for 1 min (1×), 95 °C for 15 s, and 60 °C for 30 s (40×), and finally a melting curve step at 60–95 °C was performed to check for primer specificity.

### 4.4. Phytohormone Measurements

To measure phytohormone concentrations, five leaves of infected and mock-inoculated plants were harvested at 8 dpi. The midvein was removed before leaves were weighed and immediately frozen in liquid nitrogen. Phytohormone analysis was performed via liquid chromatography-coupled tandem mass spectrometry (LC-MS/MS) as described [42].

## 5. Conclusions

We found clear indications of active plant defense responses on maize against both *formae speciales* of *S. reilianum*. A major difference in the plant response to the two *formae speciales* is the capacity of SRZ to manipulate the phytohormone levels of maize, which SRS does not seem to do. The modulation of phytohormone levels is a relatively mild adaptation to the host that presumably allows SRZ to successfully complete its life cycle on maize. In other systems, the adaptation to the host is much more convincing. For example, the maize smut fungus *U. maydis* can completely downregulate initially induced defense responses upon penetration [20] and is therefore very well adapted to its host plant. Likewise, *S. reilianum* f. sp. *reilianum* does not show severe plant defense responses after successful host colonization [18], which suggests that SRS can downregulate initially induced defense responses of sorghum. This indicates that SRS is as well adapted to sorghum as *U. maydis* is to maize. Therefore, we found clear indications that maize is not the natural host of *S. reilianum*, and that adaptation to maize is a relatively recent host jump event.

## Figures and Tables

**Figure 1 ijms-24-15604-f001:**
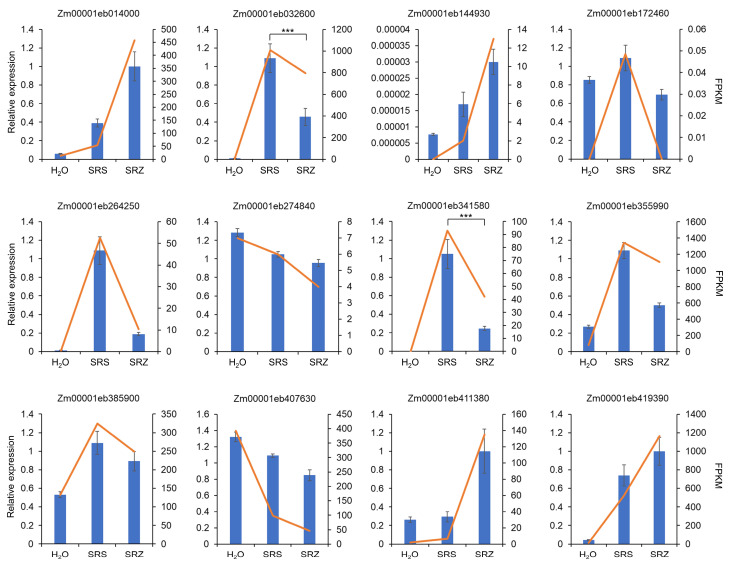
Comparison of maize gene expression of randomly selected and SA reporter genes using RT-qPCR and RNA sequencing. Samples were isolated from maize seedlings inoculated with water (H_2_O), with mating-compatible strains of *S. reilianum* f. sp. *reilianum* (SRS), or with mating-compatible strains of *S. reilianum* f. sp. *zeae* (SRZ). RT-qPCR results were first normalized to the sample of the highest expression between samples and then to the normalized expression value of the glyceraldehyde dehydrogenase (GAPDH; Zm00001eb173410) gene. Values represent means of three technical replicates each of three biological replicates (blue bars, left Y-axis). Error bars represent SEM. Sequencing results are represented by FPKM values (orange line, right Y-axis). Genes are ordered by gene number. The gene Zm00001eb172460 was barely detected with RNA sequencing only in the sample Zm-SRS, suggesting higher sensitivity of the RT-qPCR measurements. SA reporter genes PR1 and PR5 are labeled as Zm00001eb341580 and Zm00001eb032600, respectively. For these, statistical differences between SRS- and SRZ-inoculated samples were assessed using Student’s *t*-test (***, *p* < 0.001).

**Figure 2 ijms-24-15604-f002:**
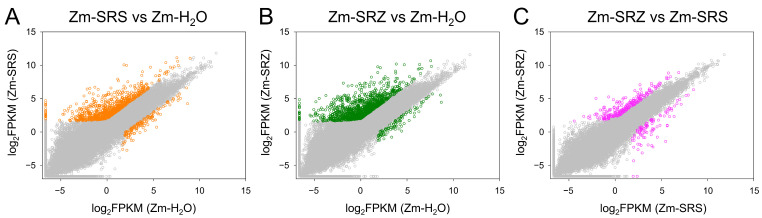
Pairwise differential expression analysis of maize genes. Maize plants were either inoculated with water (Zm-H_2_O), with two mating-compatible strains of *S. reilianum* f. sp. *reilianum* (Zm-SRS), or with two mating-compatible strains of *S. reilianum* f. sp. *zeae* (Zm-SRZ). RNA sequencing reads were mapped to the maize genome and FPKM values were compared. Differentially expressed genes are colored. Depicted are the comparisons of FPKM values of Zm-SRS with Zm-H_2_O (**A**), Zm-SRZ with Zm-H_2_O (**B**), and Zm-SRS with Zm-SRZ (**C**).

**Figure 3 ijms-24-15604-f003:**
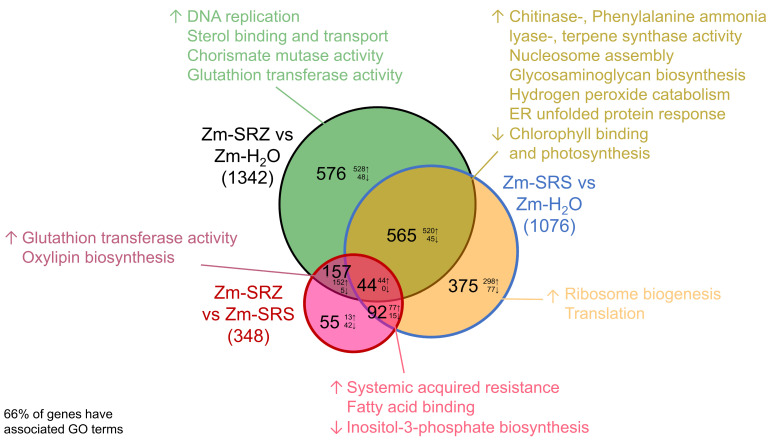
Summary of differentially expressed maize genes and their most prominent associated enriched GO terms. In the VENN diagram, the total number of differentially expressed genes is given for the respective comparisons of Zm-SRZ vs. Zm-H_2_O (black circle), Zm-SRS vs. Zm-H_2_O (blue circle), and Zm-SRZ vs. Zm-SRS (red circle). Numbers of up- (↑) and down- (↓) regulated genes are also given. The most relevant GO terms associated with each of the analyzed gene sets are given in the respective color of the VENN diagram.

**Figure 4 ijms-24-15604-f004:**
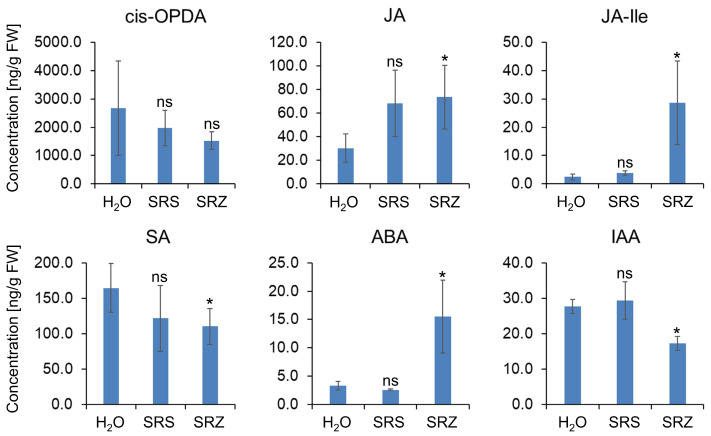
Phytohormone concentrations in maize leaf blades. Maize seedlings were inoculated with water (H_2_O), a mixture of mating-compatible strains of *S. reilianum* f. sp. *reilianum* (SRS), or a mixture of mating-compatible strains of *S. reilianum* f. sp. *zeae* (SRZ). Leaf samples were collected at 8 dpi, and the concentrations of selected phytohormones were measured using liquid chromatography-coupled tandem mass spectrometry (LC-MS/MS). Concentrations are given in ng per ng fresh weight (FW) and are shown as mean of up to five samples. Error bars show standard deviation. Significance analysis was performed relative to water-inoculated samples via Student’s *t*-test (*, *p* < 0.05; ns, not significant).

**Table 1 ijms-24-15604-t001:** Mapping statistics.

Sample	Total Reads	Uniquely Mapped Reads (% ^1^)	Unmapped Reads (% ^1^)
Zm-H_2_O	51,290,554	45,811,710 (89.3)	5,478,844 (10.7)
Zm-SRS	69,785,329	61,663,182 (88.4)	8,122,147 (11.6)
Zm-SRZ	40,313,666	35,178,627 (87.3)	5,135,039 (12.7)

^1^ Percent of total reads.

**Table 2 ijms-24-15604-t002:** Oligonucleotides used in this study.

Primer	Description	Sequence	Primer Efficiency (%)
oLD227	FP_Zm00001eb341580	GAACTCGCCTCAAGACTACC	92.6
oLD228	RP_Zm00001eb341580	TACTTCTCCGCGAACTGC	
oLD231	FP_Zm00001eb274840	ACCTACAACAGCCTGATGG	95.8
oLD232	RP_Zm00001eb274840	GCAGAACCCGTTTATGACC	
oLD235	FP_Zm00001eb144930	TTCCATCTGATTCGATCGAG	93.3
oLD236	RP_Zm00001eb144930	CACATTATTATTGGGAAACCAAC	
oSC105	FP_Zm00001eb014000	GCGTCAGGCAGTTCAACTTC	91.0
oSC106	RP_Zm00001eb014000	CCTTGGCGATCTCGTCCTTC	
oSC113	FP_Zm00001eb355990	ACGCCAAGAAGGTGATCCTC	96.7
oSC114	RP_Zm00001eb355990	CGACGATGTCGACGAAGATG	
oSC115	FP_Zm00001eb411380	TGGAGGCTGCCTTAAATGAC	99.3
oSC116	RP_Zm00001eb411380	TGTAGCGCGTGCAGTTATTG	
oSC117	FP_Zm00001eb032600	GCCAGGACTTCTACGACATC	91.1
oSC118	RP_Zm00001eb032600	GGCAGAAGGTGACTTGGTAG	
oSC121	FP_Zm00001eb172460	CATGGCCGTCATCACATGAG	99.6
oSC122	RP_Zm00001eb172460	AGTTGGTGCAGCGATGAG	
oSC123	FP_Zm00001eb264250	ACTACCCGCTTATGGTCTCC	92.9
oSC124	RP_Zm00001eb264250	ACTACTCCACGGGCAAACTC	
oSC127	FP_Zm00001eb419390	TGATACTCGTCGGCACTCTG	91.0
oSC128	RP_Zm00001eb419390	CGTTGACCGACACGTCATTG	
oSC129	FP_Zm00001eb407630	ATGCTGGCACGGAGTACAAG	97.6
oSC130	RP_Zm00001eb407630	TCCGTAAGCGCGTTTGTTGG	
oSC131	FP_Zm00001eb173410	CCATCACTGCCACACAGAAAAC	93.7
oSC132	RP_Zm00001eb173410	AGGAACACGGAAGGACATACCAG	
oSC133	FP_Zm00001eb385900	TGGGCCTACTGGTCTTACTACTGA	93.5
oSC134	RP_Zm00001eb385900	ACATACCCACGCTTCAGATCCT	

## Data Availability

The transcriptome dataset is available at the NCBI Sequence Read Archive under the BioProject ID PRJNA1022446 with the BioSample accessions SAMN37607526, SAMN37607527, and SAMN37607528.

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
