# Peer review of "Plant Responses of Maize to Two *formae speciales* of *Sporisorium reilianum* Support Recent Fungal Host Jump"

_ijms, 2023, doi:10.3390/ijms242115604_

Round 1
Reviewer 1 Report
Comments and Suggestions for Authors
The manuscript submitted by Dittiger et al. reports the transcriptional changes of maize plants infected with compatible fungal pathogen Sporisorium reilianum f.sp. zeae and incompatible S. reilianum f.sp. reilianum, of which host is sorghum. They considered that maize pathogen may arise from sorghum pathogen by host jump. The data indicates the downregulation of genes related to salicylic acid-mediated defense responses. Supporting this, the authors nicely showed the reduction of salicylic acid and further elevation of jasmonic acid. Although the connection between data and host jump is not clear for me, the manuscript, in my view, is already in good shape and its quality justifies immediate publication. I point out just a few minor points.
Line 35: Magnaporthe oryzae should be italic.
Line 206: de-novo should be italic.
Author Response
Thank you very much for the positive comments. We corrected the minor formatting errors and added a Conclusions chapter to clarify the connection between our data and our statement regarding a recent host jump event.
Reviewer 2 Report
Comments and Suggestions for Authors
The introduction must be current to give an idea of the topic in recent years, only one reference is from the last 5 years. On the other hand, in the introduction, results are not included; that is in the abstract and it is never written in the first person.
In general the redaction of the article must be revised. The materials and methods are incomplete, it is not shown where the experiment was carried out, where the pathogen used came from, what variety of corn was used, how the design of the experiment was, etc.
The results include data of materials and methods as well as discussion.
Conclusions not shown.

Author Response
Thank you for your critical comments and for supplying an annotated version of the manuscript to make your point transparent to us.
We followed most of your suggestions. In particular, we added more detailed information to the materials and methods part, we cleaned the results part of unnecessary technical details and added a conclusions chapter. We checked the introduction but cannot find more suitable more recent literature for our statements. Nevertheless, we feel that the introduction presents a current description of the scientific setting of this research. As is usual with scientific manuscripts, our introduction ends with a summary of the main findings.
In the debate of whether to use passive voice or active voice in scientific writing, we deliberately chose the latter because in our opinion it describes more honestly WHO did the experiments or drew the conclusions. It also allows for more concise language. We therefore kept our description of the findings in active voice.